# Exposure to trauma in pregnant women and its association with previous perinatal complications, IPV and antenatal service satisfaction in rural Ethiopia: a cross-sectional facility-based study

Raquel Catalao[1], Lelina Kebede[2], Adiyam Mulushoa[3], Tigist Eshetu[3], Girmay Medhin[4], Ahmed Abdella[5], Atalay Alem[6], Roxanne C. Keynejad[7], Jane Sandall[8], Louise M. Howard[7], Martin Prince[9,10], Charlotte Hanlon[3,6,11] *

1 Department of Psychological Medicine, Institute of Psychiatry, Psychology and Neuroscience, King´s College London, United Kingdom, 2 St Barnabas Hospital, Bronx, New York, United States of America, 3 Centre for Innovative Drug Development and Therapeutic Trials for Africa (CDT Africa), College of Health Sciences, Addis Ababa University, Addis Ababa, Ethiopia, 4 Aklilu-Lema Institute of Pathobiology, Addis Ababa University, Addis Ababa, Ethiopia, 5 Department of Obstetrics and Gynaecology, School of Medicine, College of Health Sciences, Addis Ababa University, Addis Ababa, Ethiopia, 6 Department of Psychiatry, WHO Collaborating Centre for Mental Health Research and Capacity Building, School of Medicine, College of Health Sciences, Addis Ababa University, Addis Ababa, Ethiopia, 7 Section of Women's Mental Health, Department of Health Service and Population Research, Institute of Psychiatry, Psychology and Neuroscience, King's College London, London, United Kingdom, 8 Department of Women and Children's Health, School of Life and Population Sciences, King's College, London, United Kingdom, 9 Centre for Global Mental Health, Health Service and Population Research Department, Institute of Psychiatry, Psychology and Neuroscience, King's College London, London, United Kingdom, 10 King's Global Health Institute, King's College London, London, United Kingdom, 11 Division of Psychiatry, Centre for Clinical Brain Sciences, University of Edinburgh, Edinburgh, Scotland, United Kingdom

* chanlon@ed.ac.uk

## Abstract

### Background

We aimed to describe the prevalence of exposure to traumatic events and post-traumatic stress disorder (PTSD) in pregnant women attending antenatal care (ANC) in rural Ethiopia. We hypothesised that antenatal PTSD symptoms would be associated with previous obstetric complications and intimate partner violence (IPV) and impact negatively on women´s satisfaction with ANC.

### Methods

The design was a facility-based cross-sectional study in primary health centres providing ANC in southern Ethiopia. Trauma events were assessed using the Life Events Checklist (LEC) and PTSD checklist for DSM-5 (PCL-5). Previous obstetric complications were extracted from clinical records. IPV was measured using the 'Non-Graphic Language' screening test and ANC satisfaction was measured using a locally

**Data availability statement:** Minimal anonymised dataset allowing replication of study findings is located in the OSF repository: https://doi.org/10.17605/OSF.IO/7S6WV.

**Funding:** The research underpinning the findings presented in this paper was funded by the National Institute of Health and Care Research (NIHR) Global Health Research Unit on Health System Strengthening in Sub-Saharan Africa (ASSET), King's College London (GHRU 16/136/54) using UK aid from the UK Government. The views expressed in this publication are those of the authors and not necessarily those of the NHS, the National Institute for Health and Care Research or the Department of Health and Social Care, England.

**Competing interests:** The authors have declared no competing interests exist.

validated adapted version of the Mental Health Service Satisfaction Scale. Generalized linear mixed-effects regression models were used to calculate prevalence ratios between PTSD, IPV and ANC satisfaction.

## Results

Out of 2079 interviewed women, 52.3% (n = 1,087) reported one or more traumatic life events on the LEC. Physical assault was the most common traumatic event experienced (n = 485; 23.3%) and witnessed (n = 1,176; 56.6%) but only 289 (13.9%) screened positive for IPV. One hundred and six women (5.1%) met DSM-5 criteria for PTSD. Women meeting diagnostic criteria for PTSD had five times increased prevalence of IPV in their current pregnancy [prevalence ratio (PR) 4.34, 95%CI 3.01–6.30; p < 0.001]. Only twenty-six women had a record of previous obstetric complications (0.01%). Overall, women with PTSD reported less satisfaction with antenatal care.

## Conclusions

Despite high exposure to traumatic life events, particularly physical violence, among pregnant women attending ANC in Southern Ethiopia, the prevalence of PTSD is relatively low. Previous obstetric complications and IPV were under-reported, relative to known prevalence estimates. Our study highlights the challenges of detection of psychosocial needs in the ANC setting and the need for targeted interventions to support women's disclosure of difficulties in maternity care settings.

## Introduction

Despite the high prevalence and adverse impacts of mental health conditions (MHCs) during the perinatal period (from conception up to two years post-partum), maternal health programmes continue to neglect women's mental health needs [1]. In low- and middle-income countries (LMICs), it is estimated 1 in 4 women experience depression in the perinatal period [2]. The prevalence of perinatal depressive and anxiety symptoms is higher among the most marginalized women, with least access to health and social care [3]. Beyond adverse birth outcomes, perinatal MHCs have transgenerational effects, negatively impacting children's health and development [4]. Inadequate recognition and low availability of treatment of perinatal MHCs in LMICs results in women remaining undiagnosed and untreated, impacting their quality of life and the physical and emotional health of their children and families [5].

Most research on perinatal MHCs in Africa has focused on common mental disorders (CMDs), namely depressive and anxiety disorders [6]. Little is known about the prevalence of perinatal post-traumatic stress disorder (PTSD) in LMICs and its consequences for women's health and future pregnancies. A systematic review of studies mostly conducted in high income countries (HICs), reported PTSD prevalence of 3.3% antenatally and 4.0% postpartum [7]. However, the prevalence of PTSD was higher in women with a history of childhood maltreatment: 18.9% antenatally and

18.5% postpartum [7]. Antenatal PTSD has been reported following traumatic life events such as accidents, interpersonal violence and natural disasters, while birth trauma is a common precipitant for postpartum PTSD [8]. Disrespectful and abusive care during childbirth is widespread in Ethiopia [9], as in other countries. In HICs, there is growing awareness of the needs of women exposed to trauma, the potential for re-traumatisation in maternity care and the requirement for therapeutic relationships and trauma-informed care [10]. Obstetric complications and IPV are commoner among women in LMICs [11] but few studies have investigated the associations between trauma exposure, PTSD, and ANC satisfaction among pregnant women in low resource settings. There is growing evidence from both HICs and LMICs that person-centered maternity care, which is respectful and responsive, supports women´s autonomy and engages the women in their care leads to improved outcomes [12].

In this study we aimed to describe the prevalence of exposure to traumatic events and PTSD symptoms in pregnant women attending ANC in rural Ethiopia and their associations with previous obstetric complications, IPV and ANC satisfaction. We hypothesised that antenatal PTSD symptoms would be associated with previous obstetric complications and current IPV and would impact negatively on women´s satisfaction with antenatal care received.

## Materials and methods

### Study design

We conducted secondary analysis of data collected in a facility-based cross-sectional study that formed part of a larger study (ASSET) of health system strengthening in sub-Saharan Africa [13]. ASSET was a multi-country consortium in Ethiopia, Sierra Leone, South Africa, and Zimbabwe which aimed to improve access to high quality care across three health care platforms: integrated continuing care for non-communicable diseases including mental health conditions (NCD/MH), maternal and newborn care, and surgical care. The data for this analysis was part of the maternal and newborn care platform in Ethiopia [14].

### Study setting

The study was conducted in eight purposively selected health centres in Meskan and Sodo districts, in the Gurage Zone of the Southern Nations, Nationalities and People's Region (SNNPR) of Ethiopia, recently renamed Central Ethiopia Regional State. According to census projections, in 2019 SNNPR had a total population of 20.1 million people, of whom 52.7% were women aged 15–49 years. The median age in the region is 20 years, 84.3% of the population live in rural areas and the total fertility rate is 3.8 births per woman [15].

### Study population

A total of 2079 pregnant women attending health centers consecutively for ANC, who were able to converse in Amharic and provide written informed consent, were included in this study. Acutely ill women who required emergency medical attention were excluded. Data collection took place from 18th July 2019 to 9th January 2020. Women were reimbursed the equivalent of $2 for their time: a locally appropriate fee.

### Data collection

Data collection was carried out in a private room within the health facility. All data collectors were female and comprised both lay data collectors (with a minimum of high school education) and clinical data collectors (with a minimum qualification of a diploma in nursing). Data were collected with electronic tablets using Open Data Kit (ODK) software [16]. Trained lay data collectors administered fully structured questionnaires. Research nurses extracted information from medical records for each woman using a bespoke template developed for the study [14].

## Measures

**Clinical characteristics of pregnancy.** Data on clinical characteristics of the current and any past pregnancies were collected from clinical records using structured forms. Our exposure of interest was previous obstetric complications, defined as one or more of the following during a previous pregnancy or delivery: spontaneous/induced abortion, ectopic pregnancy/ ruptured ectopic pregnancy, hospital admission for pre-eclampsia/ eclampsia, pre-term labour, intra-uterine foetal death, obstructed labour; fistula, Instrumental delivery, delivery by Caesarean section, postpartum haemorrhage, previous baby with a congenital anomaly, and previous baby born weighing less than 2500g or more than 4500g.

**Trauma exposure and symptoms.** Exposure to traumatic events and trauma symptoms were assessed using the Life Events Checklist (LEC) [17] and PTSD checklist for DSM-5 (PCL-5) [18], translated into Amharic and adapted for the rural Ethiopia context [19]. In the LEC, women were asked if they had experienced traumatic life events, witnessed them or learned about them through social interactions or their occupation. The PCL-5 checklist consists of 20 statements rated from "not at all" (score 0) to "extremely" (score 4). Each item rated 2 ("moderately") or higher was considered endorsement of a symptom. PCL-5 divides PTSD symptoms, with questions 1−5 reflecting criterion B (intrusion) symptoms, questions 6−7 reflecting criterion C (avoidance) symptoms, questions 8−14 reflecting criterion D (negative alterations in cognitions and mood) symptoms, and questions 15−20 reflecting criterion E (alterations in arousal and reactivity) symptoms. Women met DSM-5 criteria for ´probable PTSD´ if they scored 31 or above on PCL-5 and endorsed at least one criterion B, C, D or E symptom.

**Depressive and anxiety symptoms.** Depressive symptoms were measured using the Patient Health Questionnaire (PHQ-9). In a previous validation study of PHQ-9 [20] in health centres in the study area, a cut-off of 5 or more provided the best discrimination between people with and without major depressive disorder but had low positive predictive value (<25%). We categorised cases as 'probable depression' if the individual scored 5 or more on the PHQ-9 and reported that it was very difficult or extremely difficult for them to function in daily life. This accorded with international diagnostic criteria for depressive disorder which require impaired functioning alongside symptoms. Anxiety symptoms were measured using the generalised anxiety disorder scale (GAD-7) [21]. Women were categorised as having low anxiety symptoms if they scored 5–9, moderate anxiety symptoms if they scored 10–14 and severe anxiety symptoms if they scored 15 or above.

**Intimate Partner Violence (IPV).** IPV was measured using the 'non-graphic language' (NGL) IPV screening test [22] based on the Conflict Tactics Scale-Revised [23]. The NGL scale comprises 5 questions on experience of physical or psychological violence with an intimate partner. It was previously found to be acceptable in the study population, where it had convergent validity with the more extensive World Health Organization (WHO) IPV questionnaire [24]. We categorised participants scoring 2 or more on questions 1 (working out arguments), 3 (partner treatment) or 4 (feeling safe) as experiencing marital discord and possible IPV.

**Service satisfaction.** Satisfaction with ANC was measured using the adapted version of the Mental Health Service Satisfaction scale, validated for Ethiopian health centres and used in the ASSET diagnostic phase [25]. The modified MHSS scale had 21 items, each with four Likert response categories (1: strongly disagree, 2: disagree, 3: agree and 4: strongly agree). Most items assessed generic aspects of outpatient care relevant to the context, such as health worker communication, waiting times, privacy, usefulness of care, cleanliness of the facility. For this study we analysed total satisfaction scores and responses to all specific questions. We dichotomised results for individual items into satisfied (answering 'agree' or 'strongly agree') and non-satisfied (answering 'disagree' or 'strongly disagree').

**Covariates.** Sociodemographic characteristics including age, marital status, place of residence, educational level and religion was gathered using structured measures in an interview format by the same lay data collectors. Social support was assessed using the Oslo Social Support scale used previously in the study setting and found to have convergent validity [26].

### Ethical considerations

We obtained ethical approval from the Addis Ababa University College of Health Sciences Institutional Review Board (Reference number: 028/18/Psy) and King's College London Research Ethics Committee (Reference number: HR-17/18–6570). The study was conducted according to the Declaration of Helsinki. All participants provided written informed consent.

### Statistical analysis

We used descriptive statistics to calculate socio-demographic, clinical characteristics and satisfaction scores for the total sample and those participants meeting criteria for probable PTSD. We used Chi-squared and Fisher exact tests to compare socio-demographic characteristics and previous obstetric complications in those with and without PTSD. To account for the effect of the number of data collection days in each health centre and average ANC attendance per health centre per day (calculated over a 6-month period), we included these variables and used health centre as the clustering variable in all the following regression analyses. Complete case analyses were performed. We used Poisson mixed-effect regression to calculate prevalence ratios of co-morbid depression and anxiety in participants meeting criteria for PTSD. We used a univariate and multivariate Poisson mixed-effects regression model, adjusted for *a priori* identified confounders: parity, education, place of residence and age, to calculate prevalence ratios between PTSD and IPV. We performed a univariate and multivariate linear mixed-effects regression test adjusted for the same confounders, to test the association between PTSD detection and ANC satisfaction. We conducted a univariate and multivariate logistic mixed-effects regression model for the association between PTSD and dichotomised satisfaction on each MHSS item. We tested differences in responses to each MHSS item between those with and without PTSD using Fisher exact tests.

We also conducted linear mixed-effects regression tests of association between PCL-5 total scores and co-morbid depression, anxiety, probable IPV and total ANC satisfaction score.

## Results

### Socio-demographic characteristics

See Table 1. Out of 2079 participants, 1,024 (49.3%) were attending their first ANC appointment. Participants' mean age was 26.0 years (standard deviation (SD) 4.9; range: 18–50 years). The majority of women (n = 2013; 96.8%) were married and had some formal education (n = 1,468; 70.6%).

### Current and previous pregnancy characteristics

Most participants were in their second trimester (54.3%, n = 1,120) of a wanted pregnancy (78.5%, n = 1,632). Median parity was 1 (interquartile range: 0–3); for 28.7% of women (n = 595) this was their first pregnancy. A history of co-morbid physical health problems was documented in the clinical records of four (n = 4) participants.

A total of 26 (0.01%) women had previous obstetric complications documented in their clinical records: recurrent miscarriage (n = 15), stillbirth (n = 12), previous Caesarean section (n = 4), hospitalisation for pre-eclampsia/eclampsia (n = 2), fistula (n = 1), and previous post-partum haemorrhage (n = 1). Ectopic pregnancy, assisted delivery, previous baby with a congenital anomaly, and previous baby born weighing less than 2500g or more than 4500g were not documented for any participants.

### Trauma exposure and PTSD symptoms

A total of 1,087 (52.3%) women reported one or more traumatic life events on the LEC (range: 0–10; see Fig 1). Twenty-seven percent of women (n = 551) reported one traumatic event and 13.2% (n = 275) reported three or more traumatic life events. Physical assault was the most common traumatic event experienced (n = 485; 23.3%) and witnessed (n = 1,176; 56.6%), whereas sexual violence was infrequently reported; 2.7% (n = 57) reported sexual assault and 1.9% (n = 40) reported other unwanted sexual experiences (Table 2). Direct exposure to armed combat was reported by five

**Table 1. Participant socio-demographic characteristics.**

| Socio-demographic characteristics | Total sample n (%) | Probable PTSD[1] n = 106 n (%) | Without probable PTSD n = 1,973 n (%) | Statistical test for significance |
|---|---|---|---|---|
| **Age (years)** | | | | |
| 18-24 | 751 (36.1%) | 26 (24.5%) | 725 (36.7%) | Chi-squared test for trend $X^2$ (1, 2,079) = 3.75 p = 0.0527 |
| 25-34 | 1144 (55.0%) | 71 (67.0%) | 1,073 (54.4%) | |
| 35-50 | 184 (8.9%) | 9 (8.5%) | 175 (8.9%) | |
| **Education** | | | | |
| No formal education | 611 (29.4%) | 25 (23.6%) | 586 (29.7%) | Chi-squared test for trend $X^2$ (1, 2,079) = 0.05 p = 0.8225 |
| Primary education | 1119 (53.8%) | 69 (65.1%) | 1,050 (53.2%) | |
| Secondary education | 290 (14.0%) | 11 (10.4%) | 279 (14.1%) | |
| Post-secondary | 59 (2.8%) | 1 (0.9%) | 58 (2.9%) | |
| **Residence** | | | | |
| Rural | 1115 (53.6%) | 52 (49.1%) | 1,063 (53.9%) | Chi-squared test $X^2$ (1, 2,079) = 0.94, p = 0.332 |
| **Marital Status** | | | | |
| Married | 2013 (96.8%) | 1,912 (96.9%) | 101 (95.3%) | Fisher´s exact test p = 0.271 |
| Single | 37 (1.8%) | 37 (1.8%) | 2 (1.9%) | |
| Separated, divorced or widowed | 29 (1.4%) | 26 (1.3%) | 3 (2.8%) | |
| **Parity (n = 2,076)** | | | | |
| 0 | 595 (28.7%) | 29 (27.4%) | 566 (28.7%) | Chi-squared test for trend $X^2$ (1, 2,076) = 0.003 p = 0.8684 |
| 1 | 485 (23.4%) | 27 (25.5%) | 458 (23.2%) | |
| 2-4 | 777 (37.4%) | 41 (38.7%) | 736 (37.4%) | |
| 5 or more | 219 (10.6%) | 9 (8.5%) | 210/10.7%) | |
| **Pregnancy intention** | | | | |
| Wanted | 1632 (78.5%) | 76 (71.7%) | 1,556 (78.9%) | Fisher´s exact test p = 0.094 |
| Initially unwanted, now wanted | 409 (19.7%) | 4 (3.8%) | 34 (1.7%) | |
| Unwanted | 38 (1.8%) | 26 (24.5%) | 383 (19.4%) | |
| **Social support** | | | | |
| Poor | 533 (25.6%) | 510 (25.9%) | 23 (21.7%) | Chi-squared test for trend $X^2$ (1, 2,079) = 0.39 p = 0.5300 |
| Intermediate | 1023 (49.2%) | 958 (48.5%) | 65 (61.3%) | |
| Strong | 523 (25.2%) | 505 (25.6%) | 18 (17.0%) | |

1 Using DSM-5 criteria.

(0.2%) women and witnessed by 90 (4.3%). Experience of a life-threatening injury or illness or severe suffering were reported by 567 women (27.3%). One hundred and six women (5.1%) met DSM-5 criteria for PTSD ('probable PTSD').

There were no significant differences in education level, residence, marital status and parity between women with and without probable PTSD (Table 1). Women meeting diagnostic criteria for PTSD were more likely to be aged 25 to 34 years old rather than younger and to report lower levels of social support than women not meeting criteria for PTSD.

Forty-four percent (n = 47) of those with probable PTSD also experienced moderate to severe anxiety symptoms compared to 1.5% (n = 29) of women without PTSD. Nearly a quarter of participants with probable PTSD experienced co-morbid probable depression (n = 25; 23.6%), compared to 4.3% (n = 85) of those without PTSD. Women with probable PTSD were significantly more likely to experience co-morbid probable depression (prevalence ratio (PR): 5.50, 95% CI: 3.99–7.59, p < 0.001) and severe anxiety symptoms (PR: 20.80, 95% CI: 13.87–31.20, p < 0.001) in analyses using cut-off score of 31 on PCL-5 as well as in analyses using total PCL-5 score as continuous variable (S1 Table).

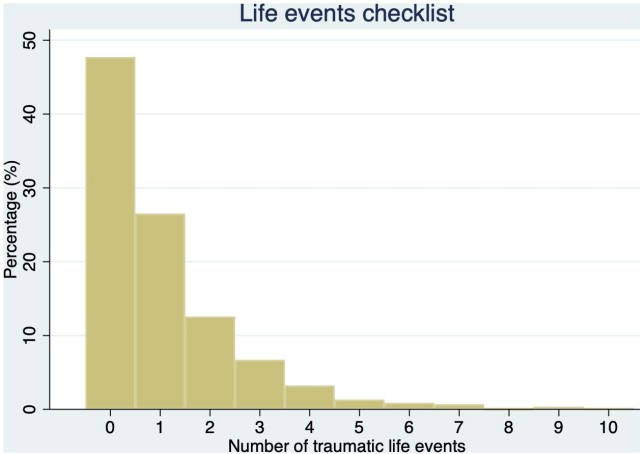

**Fig 1. Life events checklist.**

**Table 2. Types of Traumatic Life Events directly experienced and witnessed by women.**

| Life event | Directly experienced (n, %) | Witnessed (n, %) |
|---|---|---|
| Physical assault | 485 (23.3%) | 1,176 (56.6%) |
| Life threatening injury/illness | 396 (19.1%) | 846 (40.7%) |
| Natural Disaster | 221 (10.6%) | 556 (26.7%) |
| Fire | 182 (8.8%) | 809 (38.9%) |
| Other very stressful event | 175 (8.4%) | 288 (13.8%) |
| Severe human suffering | 171 (8.2%) | 646 (31.1%) |
| Motor vehicle accident | 99 (4.8%) | 520 (25.0%) |
| Captivity | 85 (4.1%) | 391 (18.8%) |
| Sudden unexpected death of loved one | 80 (3.8%) | 661 (31.8%) |
| Other serious accident | 64 (3.1%) | 231 (11.1%) |
| Sexual assault | 57 (2.7%) | 110 (5.3%) |
| Assault with a weapon | 51 (2.5%) | 312 (15.0%) |
| Sudden violent death | 50 (2.4%) | 495 (23.8%) |
| Other unwanted sexual experience | 40 (1.9%) | 58 (2.8%) |
| Caused serious injury/ death of another | 18 (0.9%) | 155 (7.5%) |
| Exposure to toxic substance | 11 (0.5%) | 41 (2.0%) |
| Combat | 5 (0.2%) | 90 (4.3%) |

## Associations between previous obstetric complications and trauma symptoms

Very few participants' clinical records documented obstetric complications in a previous pregnancy (n = 24; 0.01% in women not meeting criteria for PTSD and n = 2 in women meeting criteria for PTSD); the association was not statistically significant (p = 0.502).

## Association between trauma symptoms and IPV

A larger proportion of women with probable PTSD screened positive for probable IPV (41.5%; n = 44) compared to women without PTSD (n = 245; 12.4%). Women meeting diagnostic criteria for PTSD had four times greater risk of reporting IPV in

their current pregnancy in both the unadjusted model (PR: 4.30, 95% CI: 2.25–8.21, p < 0.001), after adjustment for parity, age, education and area of residence (PR 4.20 95%CI 2.05–8.61; p < 0.001) and in our analysis using PCL-5 total score as continuous variable (supplementary S1 Table). Of those screening positive for IPV in the total sample (n = 289, 13.9%), 133 (46.0%) reported directly experiencing physical assault and 19 (6.6%) sexual assault.

## Association between trauma symptoms and service satisfaction

The mean total satisfaction with ANC score was 57.59 (SD = 8.42) for women without probable PTSD and 56.40 (SD = 6.27) in women with probable PTSD. The mean difference was 1.19 (95% CI: −0.44 to −2.82) which when accounting for clustering by health center was found to be significant in both unadjusted analyses [regression coefficient: −0.73 (95% CI: −1.35 - −0.11, p = 0.022)] and after adjustment for parity, age, education and area of residence [adjusted regression coefficient (adj RC): −0.73 (95% CI: −1.36 - −0.10, p = 0.022)].

Women meeting criteria for PTSD were more likely to report that ANC waiting times were acceptable [adjusted Odds Ratio (adj OR) =2.35, 95% CI: 1.64–3.38, p < 0.001)], that their privacy was respected (adj OR =2.94, 95% CI: 1.91–4.52, p < 0.001), that their information was kept confidential (adj OR =2.14, 95% CI: 1.61–2.85, p < 0.001) and that referral to other services was possible (adj OR =3.64, 95% CI: 1.82–13.00, p = 0.047). Participants with probable PTSD were significantly less likely to report that the health worker involved their family helpfully (adj OR =0.29, 95% CI: 0.12–0.67, p < 0.001) and that they can afford to attend the health facility for treatment (adj OR 0.60, 95% CI 0.37–0.99, p = 0.047) (Table 3). There were significant differences in satisfaction responses between women with and without probable PTSD for all ANC satisfaction items. Women fulfilling criteria for PTSD were more likely to answer "agree" rather than "strongly agree" to most items, despite similar overall satisfaction scores (Table 4).

## Discussion

Our facility-based cross-sectional study identified high levels of exposure to life-threatening injury, illness and suffering but paradoxically a low prevalence of antenatal PTSD (5.1%), similar to previous survey of PTSD symptoms conducted in Nigeria [27]. Over half the participants had directly experienced trauma, with physical violence victimisation especially common. Nearly half of those reporting physical violence screened positive for IPV. Very few women disclosed sexual trauma. There may be several reasons for nondisclosure. Stigma associated with sexual assault deters many victim-survivors from speaking about their experiences [28]. Sexual discrimination against girls is common in Ethiopia [29] and worldwide. A multiregional study conducted by World Health Organization estimated the prevalence of lifetime sexual violence among women to be 30% [30]. In the Ethiopian Demographic and Health Survey (EDHS) 2016, 10% of women aged 15–49 years were reported to have experienced sexual violence [31] Furthermore, previous studies have found that many women and girls in Ethiopia do not disclose their experience of domestic violence [32], despite known high prevalence [33]. Reasons include shame, embarrassment and fear of disclosure related consequences. Less than 15% of our sample disclosed IPV, despite the estimated prevalence of antenatal IPV in Ethiopia ranging between 21.0% [34] and 26.1% [35]. IPV has been associated with post-partum depressive symptoms in the Ethiopian context [34]. Our study found that IPV was significantly associated with antenatal PTSD symptoms.

Among participants who had experienced a previous pregnancy, less than 1% of clinical records documented obstetric complications. This is not consistent with national estimates of around 20% [36]. Prolonged labour is the leading cause of major direct obstetric morbidity in Ethiopia, accounting for 23.4%. Hypertensive disorders are the second most frequent cause of major direct maternal morbidity with a prevalence of around 11% [36]. Reasons behind low recording of obstetric complications need to be further explored. Women with previous obstetric complications may be less prepared or less likely to attend ANC in subsequent pregnancies, or feel unable to disclose previous complications to health workers, for example in case it negatively impacts their care. It is also likely health workers may not actively ask about previous complications or do so in a manner that does not promote disclosure. Unfortunately, disrespectful and abusive care during

**Table 3. Logistic mixed-effects regression models of associations between PTSD and ANC satisfaction.**

| | | Probable PTSD (n = 106) | | Not probable PTSD (n = 1,973) | | Univariate logistic mixed-effects regression model Odds ratio (OR) (95% confidence interval (CI)) | Multivariate logistic mixed-effects regression model adjusted for parity, age, education and residence Adjusted OR (95%CI) |
|---|---|---|---|---|---|---|---|
| | | Satisfied | Not Satisfied | Satisfied | Not satisfied | | |
| 1 | The health worker treated me with courtesy | 102 (96.2%) | 4 (3.8%) | 1,891 (95.8%) | 82 (4.2%) | 1.01 (0.26-4.81) | 1.09 (0.24-4.98) |
| 2 | The health worker listened to me carefully | 104 (98.1%) | 2 (1.9%) | 1,892 (95.9%) | 81 (4.1%) | 2.01 (0.62- 6.50) | 2.10 (0.58–7.53) |
| 3 | The health worker explained things to me in a way I understood | 93 (87.7%) | 13 (12.3%) | 1,693 (85.8%) | 280 (14.2%) | 1.10 (0.54- 2.28) | 1.17 (0.57- 2.40) |
| 4 | The health facility was clean | 98 (92.4%) | 8 (7.6%) | 1,882 (95.4%) | 91 (4.6%) | 0.63 (0.29-1.39) | 0.57 (0.27-1.21) |
| 5 | The waiting room was clean | 98 (92.4%) | 8 (7.6%) | 1,799 (91.2%) | 174 (8.8%) | 1.02 (0.76-1.37) | 1.04 (0.78-1.39) |
| 6 | The latrine was clean | 75 (70.8%) | 31 (29.3%) | 1,298 (65.8%) | 675 (34.2%) | 1.18 (0.83- 1.67) | 1.21 (0.88-1.71) |
| 7 | The waiting time was acceptable | 88 (83.0%) | 18 (17.0%) | 1,312 (66.5%) | 661 (33.5%) | 2.35 (1.64-3.37) | 2.35 (1.64-3.38) |
| 8 | I had enough time to discuss with health worker | 98 (92.4%) | 8 (7.6%) | 1,794 (90.9%) | 179 (9.1%) | 1.12 (0.53- 2.39) | 1.06 (0.46- 2.45) |
| 9 | I was given information in a way I understood | 94 (88.7%) | 12 (11.3%) | 1,651 (83.7%) | 322 (16.3%) | 1.46 (0.68-3.25) | 1.38 (0.60- 3.17) |
| 10 | I received helpful advice | 85 (80.2%) | 21 (19.8%) | 1,410 (71.5%) | 563 (28.5%) | 1.59 (0.92-2.77) | 1.58 (0.93-.2.68) |
| 11 | The administrative staff treated me with courtesy and respect | 102 (96.2%) | 4 (3.8%) | 1,839 (93.2%) | 134 (6.8%) | 1.65 (0.61- 4.49) | 1.76 (0.61-5.14) |
| 12 | The health worker involved my family helpfully | 20 (18.9%) | 86 (81.1%) | 1,103 (55.9%) | 870 (44.1%) | 0.29 (0.12- 0.67) | 0.29 (0.12- 0.67) |
| 13 | My privacy was respected | 98 (92.4%) | 8 (7.6%) | 1,582 (80.2%) | 391 (19.8%) | 2.91 (1.88 −4.52) | 2.94 (1.91–4.52) |
| 14 | I have the opportunity for follow up with the same health worker | 21 (19.8%) | 85 (80.2%) | 481 (24.4%) | 1,492 (75.6%) | 0.82 (0.55- 1.24) | 0.83 (0.54- 1.23) |
| 15 | My personal information is kept confidential | 89 (84.9%) | 17 (16.0%) | 1,428 (72.4%) | 545 (27.6%) | 2.07 (1.53- 2.79) | 2.14 (1.61-2.85) |
| 16 | Referral to specialist is possible | 104 (98.1%) | 2 (1.9%) | 1,847 (93.6%) | 126 (6.4%) | 3.50 (1.00-12.05) | 3.64 (1.82- 13.00) |
| 17 | It is possible to see the health worker when needed | 99 (93.4%) | 7 (6.6%) | 1,766 (89.5%) | 207 (10.5%) | 1.65 (0.54- 5.07) | 1.72 (0.53–5.52) |
| 18 | It was easy to attend the health facility | 91 (85.8%) | 15 (14.2%) | 1,705 (86.4%) | 268 (13.6%) | 0.98 (0.70- 1.39) | 0.93 (0.63- 1.36) |
| 19 | I had enough time to attend the health facility | 98 (92.4%) | 8 (7.6%) | 1,794 (90.9%) | 179 (9.1%) | 0.69 (0.42- 1.13) | 0.67 (0.40–1.13) |
| 20 | I could afford to attend the health facility for treatment | 61(57.5%) | 45 (42.5%) | 1,384 (70.2%) | 589 (29.8%) | 0.61 (0.36- 1.01) | 0.60 (0.37–0.99) |
| 21 | I would advise my family members to come to this facility for treatment if they were pregnant | 104 (98.1%) | 2 (1.9%) | 1,889 (95.7%) | 85 (4.3%) | 2.15 (0.57- 8.08) | 2.03 (0.46- 8.92) |

childbirth is widespread in Ethiopia [9], as in other countries. Reported prevalence of disrespect and abuse during childbirth in Ethiopia ranges from 21.1% to 98.9% [9]. Qualitative research has described how fear of using health facilities is a common consequence of experiencing disrespectful care [37].

Women with antenatal PTSD showed a lower overall satisfaction with ANC compared to women without antenatal PTSD but paradoxically scored certain aspects of care higher than women without PTSD. Indeed, they endorsed more

**Table 4. ANC satisfaction in women with and without probable PTSD.**

| | | Women with probable PTSD (n = 106) | | | | Women without probable PTSD (n = 1,973) | | | | Fisher exact test of differences in category of responses |
|---|---|---|---|---|---|---|---|---|---|---|
| | | Strongly agree | Agree | Disagree | Strongly Disagree | Strongly agree | Agree | Disagree | Strongly Disagree | |
| 1 | The health worker treated me with courtesy | 13 (12.3%) | 89 (84.0%) | 3 (2.8%) | 1 (0.9%) | 472 (23.9%) | 1,419 (71.9%) | 64 (3.2%) | 18 (0.9%) | P = 0.026 |
| 2 | The health worker listened to me carefully | 9 (8.5%) | 95 (89.6%) | 1 (0.9%) | 1 (0.9%) | 403 (20.4%) | 1,489 (75.5%) | 66 (3.3%) | 15 (0.8%) | P = 0.004 |
| 3 | The health worker explained things to me in a way I understood | 8 (7.6%) | 85 (80.2%) | 7 (6.6%) | 6 (5.6%) | 330 (16.7%) | 1,363 (69.1%) | 218 (11.0%) | 62 (3.1%) | P = 0.008 |
| 4 | The health facility was clean | 7 (6.6%) | 91 (85.9%) | 8 (7.5%) | 0 (0%) | 299 (15.4%) | 1,583 (80.2%) | 74 (3.5%) | 17 (0.9%) | P = 0.017 |
| 5 | The waiting room was clean | 5 (4.7%) | 93 (87.7%) | 7 (6.6%) | 1 (0.9%) | 259 (13.1%) | 1,540 (78.1%) | 10.8 (5.5%) | 66 (3.3%) | P = 0.020 |
| 6 | The latrine was clean | 4 (3.8%) | 71 (67.0%) | 25 (23.6%) | 6 (5.6%) | 216 (10.9%) | 459 (23.3%) | 1,208 (61.2%) | 90 (4.6%) | P = 0.355 |
| 7 | The waiting time was acceptable | 5 (4.8%) | 83 (78.3%) | 12 (11.3%) | 6 (5.6%) | 224 (11.4%) | 1,088 (55.1%) | 509 (25.8%) | 172 (7.7%) | P < 0.001 |
| 8 | I had enough time to discuss with health worker | 8 (7.6%) | 90 (84.9%) | 5 (4.7%) | 3 (2.8%) | 347 (17.6%) | 1,447 (73.3%) | 147 (7.5%) | 32 (1.6%) | P = 0.012 |
| 9 | I was given information in a way I understood | 8 (7.6%) | 86 (81.1%) | 9 (8.5%) | 3 (2.8%) | 298 (15.1%) | 1,353 (68.6%) | 250 (12.7%) | 72 (3.6%) | P = 0.049 |
| 10 | I received helpful advice | 7 (6.6%) | 78 73.6% | 17 (16.0%) | 4 (3.8%) | 277 (14.0%) | 1,133 (57.4%) | 422 (21.4%) | 141 (7.2%) | P = 0.009 |
| 11 | The administrative staff treated me with courtesy and respect | 6 (5.6%) | 96 (90.6) | 2 (1.9%) | 2 (1.9%) | 342 (1.9%) | 1,497 (75.9%) | 96 (4.9%) | 38 (1.9%) | P = 0.002 |
| 12 | The health worker involved my family helpfully | 4 (3.8%) | 16 (15.1%) | 75 (70.8%) | 11 (10.4%) | 244 (12.4%) | 626 (31.7%) | 770 (39.0%) | 333 (16.9%) | P < 0.001 |
| 13 | My privacy was respected | 7 (6.6%) | 91 (85.9%) | 2 (1.9%) | 6 (5.7%) | 341 (17.3%) | 1,241 (62.9%) | 197 (10.0%) | 194 (9.8%) | P < 0.001 |
| 14 | I have the opportunity for follow up with the same health worker | 1 (0.9%) | 20 (18.9%) | 70 (66.0%) | 1 (0.9%) | 148 (7.5%) | 333 (16.9%) | 945 (47.9%) | 547 (27.7%) | P < 0.001 |
| 15 | My personal information is kept confidential | 6 (5,7%) | 83 (78.3%) | 10 (9.4%) | 7 (6.6%) | 326 (16.5%) | 1,102 (55.9%) | 315 (16.0%) | 230 (11.7%) | P < 0.001 |
| 16 | Referral to specialist is possible | 9 (8.5%) | 95 (89.6%) | 0 (0%) | 2 (1.9%) | 405 (20.5%) | 1,442 (73.1%) | 60 (3.0%) | 66 (3.4%) | P = 0.001 |
| 17 | It is possible to see the health worker when needed | 8 (7.6%) | 91 (85.8%) | 5 (4.7%) | 2 (1.9%) | 358 (18.1%) | 1, 408 (71.4%) | 133 (6.7%) | 74 (3.8%) | P = 0.009 |
| 18 | It was easy to attend the health facility | 6 (5.7%) | 85 (80.2%) | 11 (10.4%) | 4 (3.8%) | 405 (20.5%) | 1,300 (65.9%) | 169 (8.6%) | 99 (5.0%) | P < 0.001 |
| 19 | I had enough time to attend the health facility | 6 (5.7%) | 83 (78.3%) | 10 (9..4%) | 7 (6.6%) | 383 (19.4%) | 1,358 (68.8%) | 97 (4.9%) | 135 (6.8%) | P < 0.001 |
| 20 | I could afford to attend the health facility for treatment | 2 (1.9%) | 59 (55.7%) | 32 (30.2%) | 13 (12.3%) | 203 (10.3%) | 1,181 (59.9%) | 379 (19.2%) | 210 (10.6%) | P < 0.001 |
| 21 | I would advise my family members to come to this facility for treatment if they were pregnant | 10 (9.4%) | 94 (88.7%) | 0 (0%) | 2 (1.9%) | 516 (26.2%) | 1,373 (69.6%) | 44 (2.2%) | 40 (2.0%) | P < 0.001 |

items regarding respect for their privacy and confidentiality. This finding should be interpreted with caution due to potential for response bias if women believe their answers may influence further care. In addition, previous studies in lower income countries have shown that lower expectations can lead to higher levels of satisfaction with care [38]. A study from Ethiopia also reported higher satisfaction with ANC care to be associated with lower levels of education [39] and women in our sample with antenatal PTSD were more likely to have lower levels of education than those without, highlighting the complex interplay between education, expectations and potential social desirability bias (the tendency to underreport socially undesirable attitudes and behaviors and to over report more desirable attributes). Women with probable PTSD were less likely than women without probable PTSD to report that the health worker engaged their family helpfully and that they could afford to attend antenatal care. As IPV was more prevalent among participants with probable PTSD, training on sensitive ways to involve family members and ensure women are supported to attend ANC is a particular priority. Women with probable PTSD were less likely to endorse "strongly agree" responses for most aspects of care provided, indicating a systematic difference in reporting and highlighting the need to develop sensitive and validated scales to adequately capture the actual experiences of women accessing care.

Our results highlight the importance of person-centred maternity care to improve outcomes and experience of care for women. Previous studies from LMICs indicate that higher ratings of ANC dignity, respect, and supportiveness are associated with fewer newborn complications and increased willingness to access ANC for future pregnancies [40]. The absence of person-centred maternity care has been associated with birth and post-partum complications [41]. In 2019, the Federal Ministry of Health of Ethiopia introduced the Ethiopian Primary Health care Clinical Guidelines (EPHCG) promoting integrated, person-centred and evidence-based care in health centres [42]. In order to promote person-centred maternity care, systemic changes are required in tandem with health worker training and supervision. The ASSET team collaborated with the Ministry of Health to adapt training resources in person-centred care for maternal care in Ethiopia [43] Beyond training, services need to empower women with greater understanding of their rights and instil trust to facilitate disclosure of trauma; however, this requires cultural changes at the organisational level, including support and supervision for health workers, manageable caseloads and conductive clinical environments [44]. Previous research in South Africa highlighted that efforts to improve detection of mental health conditions and IPV need to occur concurrently with development of pathways for referral and treatment/support for women to be confident their needs will be listened to and acted upon [44].

Despite high levels of high trauma exposure, only a small proportion of women in our sample met criteria for PTSD. Qualitative research is required to explore how women respond to and cope with traumatic life events in this context. While perinatal depression is widely researched, less is known about the impacts of PTSD and trauma exposure on woman and child outcomes [45]. Our finding of high comorbidity between perinatal PTSD, depressive symptoms, and IPV highlights the need for studies to measure a range of CMDs and aetiologically relevant social determinants. Further research into perinatal PTSD and trauma symptoms, especially their association with birth and other obstetric trauma. Such studies in representative community samples will aid the development of trauma-informed, context-appropriate interventions for women across the perinatal period.

## Strengths and limitations

Rural populations in Ethiopia face multiple barriers to care including low awareness of, and high stigma, related to mental health symptoms [46,19]. Previous studies have shown that females and those with lower level of education have higher internalised self-stigma regarding mental illness in the country [47]. Therefore, fear and stigma of sensitive topics such as IPV, trauma and mental health symptoms may have contributed to significant underreporting in our study. This is the first study in Ethiopia and one of few from LMICs, to investigate the relationships between trauma symptoms, IPV, obstetric complications and ANC satisfaction using contextually validated (for satisfaction) and adapted (trauma, IPV) measures in a large sample. Limitations include potential under detection of traumatic events, PTSD and IPV. There is a need for further studies to examine how trauma symptoms may manifest differently in this setting, the validity of cut-off points used by our

measuring tools and to develop targeted interventions to elicit sensitive disclosure. ANC satisfaction may also have been impacted by social desirability bias as well as concerns how responses may impact further care and attention is needed to confidential ways of providing women with safe spaces to candidly disclose their experiences. The satisfaction scale used was initially developed to capture satisfaction with mental health services and although it includes several of the relevant concepts for person-centered care including dignity, respect and collaboration, it may be missing other important aspects of maternity-specific care. In higher income settings, trauma-informed approaches have been increasingly cited in policy and adopted in practice as a means for reducing the negative impact of trauma experiences and supporting mental and physical health outcomes and prevent re-traumatisation [48]. Whereas several of the principles of person-centered care overlap with principles of trauma-informed care (such as trustworthiness, safety, collaboration and choice), further studies are required to understand how local policies, protocols and processes can be responsive to the needs of individuals served and how they can be empowered in their interactions with the health system. Due to the cross-sectional design of our study, we are also unable to draw casual inferences; studies with a longitudinal design would be helpful to further identify the trajectory of trauma symptoms and their impact on psychological wellbeing and ANC satisfaction over time in this setting. The low recording of obstetric complications, inconsistent with national prevalence estimates, highlights how health system strengthening initiatives need to address the quality of data recording by clinicians. Future research should explore women's, maternal health care workers' and other relevant stakeholders', including intimate partners', perspectives on high quality ANC and person-centred care and specifically how services can be adapted to ensure the necessary space is provided to facilitate sensitive disclosure of trauma, validate feelings and concerns of staff and patients, support people to make decisions and take action and involve family members in a constructive and culturally-acceptable manner. Recently, there has been close working between the Ethiopian Federal Ministry of Health and the ASSET research team to integrate the Practical Approach to Care Kit (PACK) programme, which comprises the provision of carefully designed comprehensive and integrated clinical decision support into primary health care [49] and develop training on person-centered maternal care [14]. In addition, there is ongoing research to support the development of brief psychological interventions for perinatal women with exposure to IPV that can be feasibly delivered by primary care workers [50,51].

## Conclusions

Pregnant women attending ANC in Ethiopia report frequent exposure to traumatic life events, particularly physical violence. However, the prevalence of detected probable PTSD is low. IPV is associated with perinatal PTSD symptoms. Obstetric complications and IPV were under-reported, relative to previous studies in Ethiopia. ANC satisfaction was lower in women with probable PTSD. Our study highlights the importance of person-centred approaches for improving recognition of trauma exposure, antenatal PTSD and the quality of ANC.

## Supporting information

**S1 Table. Associations between PCL-5 total score mental health disorders, IPV obstetric complications and ANC satisfaction.**
(DOCX)

**S2. Inclusivity-in-global-research-questionnaire_Catalao paper.**
(DOCX)

## Acknowledgments

CH and MP acknowledge financial support from a National Institute for Health and Care Research (NIHR) global health research group on homelessness and mental health in Africa (HOPE; NIHR134325). The views expressed in this publication are those of the authors and not necessarily those of the NHS, the National Institute for Health and Care Research

or the Department of Health and Social Care, England. CH also acknowledges funding from the Wellcome Trust through grants 222154/Z20/Z (SCOPE) and 223615/Z/21/Z (PROMISE). the funders had no role in study design, data collection and analysis, decision to publish, or preparation of the manuscript.

For the purpose of open access, the authors have applied a Creative Commons Attribution (CC BY) licence to any Author Accepted Author Manuscript version arising from this submission.

We thank all the women who kindly agreed to participate in this study.

## Author contributions

**Conceptualization:** Raquel Catalao, Lelina Kebede, Charlotte Hanlon.

**Data curation:** Raquel Catalao.

**Formal analysis:** Raquel Catalao, Lelina Kebede, Girmay Medhin.

**Funding acquisition:** Charlotte Hanlon.

**Investigation:** Adiyam Mulushoa, Tigist Eshetu, Ahmed Abdella.

**Methodology:** Raquel Catalao, Girmay Medhin, Charlotte Hanlon.

**Project administration:** Tigist Eshetu, Ahmed Abdella.

**Resources:** Tigist Eshetu.

**Supervision:** Charlotte Hanlon.

**Writing – original draft:** Raquel Catalao.

**Writing – review & editing:** Lelina Kebede, Adiyam Mulushoa, Tigist Eshetu, Girmay Medhin, Atalay Alem, Roxanne C. Keynejad, Jane Sandall, Louise M Howard, Martin Prince, Charlotte Hanlon.

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
