## [Decision Letter · Decision Letter 0]

12 May 2025

PONE-D-25-05649Exposure to trauma in pregnant women and its association with previous perinatal complications, IPV and antenatal service satisfaction in rural Ethiopia: a cross-sectional facility-based studyPLOS ONE

Dear Dr. Hanlon,

Thank you for submitting your manuscript to PLOS ONE. After careful consideration, we feel that it has merit but does not fully meet PLOS ONE’s publication criteria as it currently stands. Therefore, we invite you to submit a revised version of the manuscript that addresses the points raised during the review process.

**Two review reports have been obtained. Please revise them accordingly.**

We look forward to receiving your revised manuscript.

Kind regards,

Muhammad Haroon Stanikzai

Academic Editor

PLOS ONE

**Journal Requirements:**

1. When submitting your revision, we need you to address these additional requirements. Please ensure that your manuscript meets PLOS ONE's style requirements, including those for file naming. The PLOS ONE style templates can be found at https://journals.plos.org/plosone/s/file?id=wjVg/PLOSOne_formatting_sample_main_body.pdf and https://journals.plos.org/plosone/s/file?id=ba62/PLOSOne_formatting_sample_title_authors_affiliations.pdf 2. Please include a complete copy of PLOS’ questionnaire on inclusivity in global research in your revised manuscript. Our policy for research in this area aims to improve transparency in the reporting of research performed outside of researchers’ own country or community. The policy applies to researchers who have travelled to a different country to conduct research, research with Indigenous populations or their lands, and research on cultural artefacts. The questionnaire can also be requested at the journal’s discretion for any other submissions, even if these conditions are not met.  Please find more information on the policy and a link to download a blank copy of the questionnaire here: https://journals.plos.org/plosone/s/best-practices-in-research-reporting. Please upload a completed version of your questionnaire as Supporting Information when you resubmit your manuscript. 3. Thank you for stating the following financial disclosure: The research underpinning the findings presented in this paper was funded by the National Institute of Health and Care Research (NIHR) Global Health Research Unit on Health System Strengthening in Sub-Saharan Africa (ASSET), King’s College London (GHRU 16/136/54) using UK aid from the UK Government. CH and MP are funded by an NIHR global health research group on homelessness and mental health in Africa (HOPE; NIHR134325). The views expressed in this publication are those of the authors and not necessarily those of the NHS, the National Institute for Health and Care Research or the Department of Health and Social Care, England. CH is also funded by the Wellcome Trust through grants 222154/Z20/Z (SCOPE) and 223615/Z/21/Z (PROMISE).   Please state what role the funders took in the study.  If the funders had no role, please state: "The funders had no role in study design, data collection and analysis, decision to publish, or preparation of the manuscript." If this statement is not correct you must amend it as needed. Please include this amended Role of Funder statement in your cover letter; we will change the online submission form on your behalf. 4. When completing the data availability statement of the submission form, you indicated that you will make your data available on acceptance. We strongly recommend all authors decide on a data sharing plan before acceptance, as the process can be lengthy and hold up publication timelines. Please note that, though access restrictions are acceptable now, your entire data will need to be made freely accessible if your manuscript is accepted for publication. This policy applies to all data except where public deposition would breach compliance with the protocol approved by your research ethics board. If you are unable to adhere to our open data policy, please kindly revise your statement to explain your reasoning and we will seek the editor's input on an exemption. Please be assured that, once you have provided your new statement, the assessment of your exemption will not hold up the peer review process. 5. Please amend your authorship list in your manuscript file to include authors Dr. Raquel Catalao, Lelina Kebede, Adiyam Mulushoa and Tigist Eshetu. 6. Please upload a copy of Figure 1, to which you refer in your text on page 16. If the figure is no longer to be included as part of the submission please remove all reference to it within the text.

Reviewers' comments:

Reviewer's Responses to Questions

**Comments to the Author**

1. Is the manuscript technically sound, and do the data support the conclusions?

Reviewer #1: Yes

Reviewer #2: Yes

2. Has the statistical analysis been performed appropriately and rigorously? 

Reviewer #1: Yes

Reviewer #2: Yes

3. Have the authors made all data underlying the findings in their manuscript fully available?

Reviewer #1: Yes

Reviewer #2: No

4. Is the manuscript presented in an intelligible fashion and written in standard English?

Reviewer #1: Yes

Reviewer #2: No

5. Review Comments to the Author

**Reviewer #1:**  Thank you, the Editorial Team, for inviting me to review this manuscript on an important and timely topic. The manuscript is well written and present in rigours, impactful study that advances understanding of perinatal PSTD in LMICs. After a careful review, I offer the following comments and questions to help strengthen the clarity, contextual relevance, and implications of findings.

Introduction

1. Could the authors elaborate on why Ethiopia’s specific context makes this study particularly urgent? How might cultural stigma surrounding mental health and interpersonal violence affect women’s willingness to disclosure trauma in this setting?

2. Were trauma-informed care interventions considered in the study design. Particularly given the sensitive nature of PSTD and IPV??

3. Although the Study conducted in Ethiopia, and framed with in the broader LMIC context, only one LMIC (Nigeria) is cited. The authors could expand their discussion of gaps in perinatal PSTD research across LMICs to better contextualize the study’s contribution.

Methods

4. Please provide more details on how the PTSD Checklist for DSM-5 (PLC-5) and life event Checklist (LEC) were culturally adapted or validated for use in rural Ethiopia settings.

5. The study defines probable PTSD using a PCL-5 cut-off score of 31. While this ia a commonly used threshold, further justification or local validation data would strengthen its application to this population

6. Only 0.01 % participants were reported to have obstetric complications. Could extremely low figure reflect a limitation in data collection or recording?

7. How were lay data collectors trained to handle disclosures of trauma or IPV? Were any safeguards or referral mechanisms in place?

Results

8. The findings that PTSD positive women reported higher satisfaction with privacy and referral, yet lower overall satisfaction, is intriguing. Could the authors explore whether response bias or differing expectations influenced these responses?

9. How were potential comorbidities such as depression or anxiety accounted for in the analysis? Were these controlled for as confounding variables?

Discussion

10. Could the relatively low prevalence of PSTD observed reflect cultural coping mechanism or alternative expression of distress that may not be captured by PLC-5?

11. Given the facility-based study design, how might the exclusion of women who avoid ANC possible due to trauma or stigma have biased the finings? These high-risk groups might be underrepresented/

Conclusion

12. How do the findings inform the broader goals of the ASSET project in strengthening health systems? Are there specific programmatic or policy recommendations that emerged from this work?

**Reviewer #2:**  The topic is of high public health significance, particularly in LMICs where perinatal mental health is often neglected despite its profound intergenerational effects. The study highlights a gap in literature on perinatal PTSD in African settings, moving beyond the common focus on depression and anxiety. However, there are some suggestions which can strength the manuscript.

In my view, cross sectional design limits the ability to draw causal inferences between trauma exposure, PTSD, and ANC satisfaction. Longitudinal data would be better suited to assess the trajectory and impacts of PTSD over time. Potential reporting bias were not properly addressed s sensitive topics such as IPV and mental health symptoms were self-reported and may be underreported due to stigma or fear, especially in patriarchal rural settings. While the PCL-5 was used, the absence of clinical diagnostic interviews may have led to misclassification. The cut-off of 31, though standard, may need local validation.

The satisfaction scale was originally designed for mental health services. Its adaptation to ANC settings is not fully described—how well it captured maternity-specific concerns is unclear.

The categorization of IPV based on just three items may miss subtler or more severe forms of abuse. The scale might underestimate IPV prevalence. There is no mention of how missing data were handled, which could influence the validity of the results. The study presents valuable, methodologically sound findings that advance understanding of trauma and PTSD among pregnant women in a resource-limited context.

There is no mention of theoretical models (e.g., trauma-informed care frameworks, health-seeking behavior models) that could strengthen the interpretation of complex findings. This weakens the scholarly rigor of the discussion even it should be part of methodology.

The discussion briefly notes that PTSD prevalence was low, but it misses a deeper exploration of whether this reflects methodological limitations (e.g., time window of trauma, cultural variation in symptom reporting, or cutoff scores). The conclusion would benefit from a more nuanced interpretation of this discrepancy.

The observation that women with probable PTSD scored some aspects of ANC more positively is raised but not critically interrogated. Is this a reflection of lowered expectations, social desirability bias, or misinterpretation of the scale? These ambiguities weaken the interpretation. The authors rightly emphasize person-centred maternity care, but the discussion lacks depth about how this can be implemented effectively in the Ethiopian context. There is also limited integration of study findings to support this advocacy—what specific elements of care were deficient?

Finally, conclusion appropriately emphasizes the need for better recognition of trauma and integration of psychosocial care into ANC, but falls short in providing actionable insight or priorities for intervention or future research

6. PLOS authors have the option to publish the peer review history of their article (what does this mean? ). If published, this will include your full peer review and any attached files.

**Do you want your identity to be public for this peer review?** For information about this choice, including consent withdrawal, please see our Privacy Policy .

Reviewer #1: **Yes: ** Temesgen Anjulo Ageru

Reviewer #2: No

---

## [Author Response · Author response to Decision Letter 1]

9 Jun 2025

Response to reviewers

1. The topic is of high public health significance, particularly in LMICs where perinatal mental health is often neglected despite its profound intergenerational effects. The study highlights a gap in literature on perinatal PTSD in African settings, moving beyond the common focus on depression and anxiety. However, there are some suggestions which can strength the manuscript.

RESPONSE: Thank you for your comments to improve our manuscript.

2. In my view, cross sectional design limits the ability to draw causal inferences between trauma exposure, PTSD, and ANC satisfaction. Longitudinal data would be better suited to assess the trajectory and impacts of PTSD over time.

RESPONSE: We recognize the limitation of our cross-sectional analysis. In the section “Strengths and Limitations” we have now added the following sentence: “Due to the cross-sectional design of our study, we are also unable to draw casual inferences; studies with a longitudinal design would be helpful to further identify the trajectory of trauma symptoms and their impact on psychological wellbeing and ANC satisfaction over time in this setting.” (page 35).

3. Potential reporting bias were not properly addressed s sensitive topics such as IPV and mental health symptoms were self-reported and may be underreported due to stigma or fear, especially in patriarchal rural settings. While the PCL-5 was used, the absence of clinical diagnostic interviews may have led to misclassification. The cut-off of 31, though standard, may need local validation.

RESPONSE: We recognize the sensitivity of the topic and challenges with accurate measurement and have amended the “Strengths and Limitations” section on page 34 as follows: “Rural populations in Ethiopia face multiple barriers to care including low awareness of, and high stigma, related to mental health symptoms(46,47). Previous studies have shown that females and those with lower level of education have higher internalised self-stigma regarding mental illness in the country(48). Therefore, fear and stigma of sensitive topics such as IPV, trauma and mental health symptoms may have contributed to significant underreporting in our study. This is the first study in Ethiopia and one of few from LMICs, to investigate the relationships between trauma symptoms, IPV, obstetric complications and ANC satisfaction using contextually validated (for satisfaction) and adapted (trauma, IPV) measures in a large sample. Limitations include potential under detection of traumatic events, PTSD and IPV. There is a need for further studies to examine how trauma symptoms may manifest differently in this setting, the validity of cut-off points used by our measuring tools and to develop targeted interventions to elicit sensitive disclosure. “

We acknowledge the limitation that cut-off of 31 in the PCL-5 was not validated in this setting. In recognition of that, we used the term “probable PTSD” throughout the paper. In addition, we have now conducted further analyses using the PCL-5 total score as continuous measure which corroborate the direction of the associations found using the cut off score of 31 (supplementary table 1). To reflect the extra analyses, we added “We also conducted linear mixed-effects regression tests of association between PCL-5 total scores and co-morbid depression, anxiety, probable IPV and total ANC satisfaction score.” in Analysis Plan on page 10 and reported results on pages 17 and 18.

4. The satisfaction scale was originally designed for mental health services. Its adaptation to ANC settings is not fully described—how well it captured maternity-specific concerns is unclear.

RESPONSE: We have now included the following statement in the “Strengths and Limitations” section on page 34: “The satisfaction scale used was initially developed to capture satisfaction with mental health services and although it includes several of the relevant concepts for person-center care including dignity, respect and collaboration, it may be missing other important aspects of maternity-specific care.”

5. The categorization of IPV based on just three items may miss subtler or more severe forms of abuse. The scale might underestimate IPV prevalence.

RESPONSE: We thank the reviewer for raising this point. However, work by Zink et al showed that less direct ways of indicating IPV may be more accurate, and that the three-question combination including the domains of argument, safety, and manner of treating the responder had the best results (Zink T, Levin L, Putnam F, Beckstrom A. Accuracy of Five Domestic Violence Screening Questions With Nongraphic Language. Clinical Pediatrics. 2007;46(2):127-134. doi:10.1177/0009922806290029).

6. There is no mention of how missing data were handled, which could influence the validity of the results.

RESPONSE: We thank reviewer for raising this point. We now added the sentence “Complete case analyses were performed.” to the analysis plan section on page 10.

7. The study presents valuable, methodologically sound findings that advance understanding of trauma and PTSD among pregnant women in a resource-limited context.

RESPONSE: We thank the reviewer for the positive feedback and opportunity to address their comments.

8. There is no mention of theoretical models (e.g., trauma-informed care frameworks, health-seeking behavior models) that could strengthen the interpretation of complex findings. This weakens the scholarly rigor of the discussion even it should be part of methodology.

RESPONSE: In order to strengthen our interpretation and highlight commonalities between the endorsed model of person-centered care and trauma informed care we added the following sentence: “In higher income settings, trauma-informed approaches have been increasingly cited in policy and adopted in practice as a means for reducing the negative impact of trauma experiences and supporting mental and physical health outcomes and prevent re-traumatisation (48). Whereas several of the principles of person-centered care overlap with principles of trauma-informed care (such as trustworthiness, safety, collaboration and choice), further studies are required to understand how local policies, protocols and processes can be responsive to the needs of individuals served and how they can be empowered in their interactions with the health system.“ (pages 34 and 35).

9. The discussion briefly notes that PTSD prevalence was low, but it misses a deeper exploration of whether this reflects methodological limitations (e.g., time window of trauma, cultural variation in symptom reporting, or cutoff scores). The conclusion would benefit from a more nuanced interpretation of this discrepancy.

RESPONSE: In response to these important comments raised by the author we have modified the “Strengths and Limitations” section of our manuscript on page 34 as follows: “Rural populations in Ethiopia face multiple barriers to care including low awareness of, and high stigma, related to mental health symptoms(46,47). Previous studies have shown that females and those with lower level of education have higher internalised self-stigma regarding mental illness in the country(48). Therefore, fear and stigma of sensitive topics such as IPV, trauma and mental health symptoms may have contributed to significant underreporting in our study. This is the first study in Ethiopia and one of few from LMICs, to investigate the relationships between trauma symptoms, IPV, obstetric complications and ANC satisfaction using contextually validated (for satisfaction) and adapted (trauma, IPV) measures in a large sample. Limitations include potential under detection of traumatic events, PTSD and IPV. There is a need for further studies to examine how trauma symptoms may manifest differently in this setting, the validity of cut-off points used by our measuring tools and to develop targeted interventions to elicit sensitive disclosure. “ and “Due to the cross-sectional design of our study, we are also unable to draw casual inferences; studies with a longitudinal design would be helpful to further identify the trajectory of trauma symptoms and their impact on psychological wellbeing and ANC satisfaction over time in this setting.” on page 35. We also acknowledge the limitation that cut-off of 31 in the PCL-5 was not validated in this setting. In recognition of that, we used the term “probable PTSD” throughout the paper. In addition, we have now conducted further analyses using the PCL-5 total score as continuous measure which corroborate the direction of the associations found using the cut off score of 31 (supplementary table 1). To reflect the extra analyses, we added “We also conducted linear mixed-effects regression tests of association between PCL-5 total scores and co-morbid depression, anxiety, probable IPV and total ANC satisfaction score.” in Analysis Plan on page 10 and reported results on pages 17 and 18.

10. The observation that women with probable PTSD scored some aspects of ANC more positively is raised but not critically interrogated. Is this a reflection of lowered expectations, social desirability bias, or misinterpretation of the scale? These ambiguities weaken the interpretation.

RESPONSE: The reviewer made an important point regarding the need to further interrogate these findings. We amended the paragraph in “Discussion” in page 32 as following to provide a more nuanced interpretation: “Women with antenatal PTSD showed a lower overall satisfaction with ANC compared to women without antenatal PTSD but paradoxically scored certain aspects of care higher than women without PTSD. Indeed, they endorsed more items regarding respect for their privacy and confidentiality. This finding should be interpreted with caution due to potential for response bias if women believe their answers may influence further care. In addition, previous studies in lower income countries have shown that lower expectations can lead to higher levels of satisfaction with care(38). A study from Ethiopia also reported higher satisfaction with ANC care to be associated with lower levels of education(39) and women in our sample with antenatal PTSD were more likely to have lower levels of education than those without, highlighting the complex interplay between education, expectations and potential social desirability bias (the tendency to underreport socially undesirable attitudes and behaviors and to over report more desirable attributes). “

11. The authors rightly emphasize person-centred maternity care, but the discussion lacks depth about how this can be implemented effectively in the Ethiopian context. There is also limited integration of study findings to support this advocacy—what specific elements of care were deficient? Finally, conclusion appropriately emphasizes the need for better recognition of trauma and integration of psychosocial care into ANC, but falls short in providing actionable insight or priorities for intervention or future research.

RESPONSE: Thank you for highlighting the need to identify priorities for intervention. We have added the following paragraph to “Strengths and Limitations on page 35: ” Future research should explore women’s, maternal health care workers’ and other relevant stakeholders’, including intimate partners’, perspectives on high quality ANC and person-centred care and specifically how services can be adapted to ensure the necessary space is provided to facilitate sensitive disclosure of trauma, validate feelings and concerns of staff and patients, support people to make decisions and take action and involve family members in a constructive and culturally-acceptable manner. Recently, there has been close working between the Ethiopian Federal Ministry of Health and the ASSET research team to integrate the Practical Approach to Care Kit (PACK) programme, which comprises the provision of carefully designed comprehensive and integrated clinical decision support into primary health care (50)and develop training on person-centered maternal care(51). In addition, there is ongoing research to support the development of brief psychological interventions for perinatal women with exposure to IPV that can be feasibly delivered by primary care workers(52,53).”

We hope the current revision addresses all the raised points by reviewers and we remain committed to improve the manuscript to the highest standard by addressing the constructive feedback given.

---

## [Decision Letter · Decision Letter 1]

26 Jun 2025

PONE-D-25-05649R1Exposure to trauma in pregnant women and its association with previous perinatal complications, IPV and antenatal service satisfaction in rural Ethiopia: a cross-sectional facility-based studyPLOS ONE

Dear Dr. Hanlon,

Thank you for submitting your manuscript to PLOS ONE. After careful consideration, we feel that it has merit but does not fully meet PLOS ONE’s publication criteria as it currently stands. Therefore, we invite you to submit a revised version of the manuscript that addresses the points raised during the review process.

Thank for addressing reviewers' comments. Before this manuscript can be accepted for publication, I request some minor correction mentioned at additional editor comments section. 

We look forward to receiving your revised manuscript.

Kind regards,

Muhammad Haroon Stanikzai

Academic Editor

PLOS ONE

Journal Requirements:

Additional Editor Comments:

- Please look at previous publications at PLOS ONE and use in-text citations as per journal style.

- Please use capital letter for the word of heading and subheadings Please review the whole manuscript and table headings).

- Line 214: Please make it statistical analysis.

- The manuscript would be benefit from careful reading by authors (particularly capitalization and punctuation).

Reviewers' comments:

Reviewer's Responses to Questions

**Comments to the Author**

1. If the authors have adequately addressed your comments raised in a previous round of review and you feel that this manuscript is now acceptable for publication, you may indicate that here to bypass the “Comments to the Author” section, enter your conflict of interest statement in the “Confidential to Editor” section, and submit your "Accept" recommendation.

Reviewer #1: All comments have been addressed

2. Is the manuscript technically sound, and do the data support the conclusions?

Reviewer #1: Yes

3. Has the statistical analysis been performed appropriately and rigorously? 

Reviewer #1: Yes

4. Have the authors made all data underlying the findings in their manuscript fully available?

Reviewer #1: Yes

5. Is the manuscript presented in an intelligible fashion and written in standard English?

Reviewer #1: Yes

6. Review Comments to the Author

Reviewer #1: (No Response)

7. PLOS authors have the option to publish the peer review history of their article (what does this mean? ). If published, this will include your full peer review and any attached files.

**Do you want your identity to be public for this peer review?** For information about this choice, including consent withdrawal, please see our Privacy Policy .

Reviewer #1: **Yes: ** Temesgen Anjulo Ageru

---

## [Author Response · Author response to Decision Letter 2]

1 Jul 2025

Muhammad Haroon Stanikzai

Academic Editor

PLOS ONE

1st July 2025

Dear Dr Stanikzai,

Re: PONE-D-25-05649

Exposure to trauma in pregnant women and its association with previous perinatal complications, IPV and antenatal service satisfaction in rural Ethiopia: a cross-sectional facility-based study

Thank you for the opportunity to further review and improve our manuscript. We addressed the raised points as below:

Journal Requirements:

RESPONSE: We have now reviewed the full reference list to ensure references are complete and correct. The following changes have been implemented:

• Reference 14 which was published as pre-print at time of submission of our manuscript but has since been accepted and published has been added as full reference as follows:

14. Eshetu T, Fekadu E, Abdella A, Mulushoa A, Medhin G, Belina M, et al. Towards person-centred maternal and newborn care in Ethiopia: a mixed method study of satisfaction and experiences of care. BMC Pregnancy and Childbirth. 2025; 25(1):1–13. doi: 10.1186/s12884-024-07116-4.

• Reference 21 was updated to a more relevant paper of the validation of GAD-7 as measuring tool for anxiety symptoms.

21. Spitzer RL, Kroenke K, Williams JBW, Löwe B. A Brief Measure for Assessing Generalized Anxiety Disorder: The GAD-7. Arch Intern Med. 2006; 166(10):1092–7. doi:10.1001/archinte.166.10.1092

• Similarly. reference 24 was amended to a more relevant paper describing the use of our intimate partner violence measuring tool in the local setting:

24. Bitew T, Keynejad R, Honikman S, Sorsdahl K, Myers B, Fekadu A, et al. Stakeholder perspectives on antenatal depression and the potential for psychological intervention in rural Ethiopia: A qualitative study. BMC Pregnancy Childbirth. 2020; 20(1):1–11. doi:10.1186/s12884-020-03069-6.

Additional Editor Comments:

- Please look at previous publications at PLOS ONE and use in-text citations as per journal style.

RESPONSE: We have added in-text citations as per journal style including use of square brackets for references.

- Please use capital letter for the word of heading and subheadings Please review the whole manuscript and table headings).

RESPONSE: We have now reviewed the whole manuscript and implemented 3 levels of Headings with font size and bold formatting as specified. We have also added Fig 1 caption in the manuscript as well as specified supplementary information at end of manuscript.

- Line 214: Please make it statistical analysis.

RESPONSE: Changed as requested.

- The manuscript would be benefit from careful reading by authors (particularly capitalization and punctuation).

RESPONSE: Reviewed as requested. Minor changes were tracked in manuscript.

We hope this version meets the required criteria for publication in your journal.

Kind regards,

Professor Charlotte Hanlon

---

## [Editor Report · Decision Letter 2]

4 Jul 2025

Exposure to trauma in pregnant women and its association with previous perinatal complications, IPV and antenatal service satisfaction in rural Ethiopia: a cross-sectional facility-based study

PONE-D-25-05649R2

Dear Dr. Hanlon,

We’re pleased to inform you that your manuscript has been judged scientifically suitable for publication and will be formally accepted for publication once it meets all outstanding technical requirements.

Kind regards,

Muhammad Haroon Stanikzai

Academic Editor

PLOS ONE

Additional Editor Comments (optional):

Thank you for addressing reviewers' comments.
---

## [Editor Report · Acceptance letter]

PONE-D-25-05649R2

PLOS ONE

Dear Dr. Hanlon,

I'm pleased to inform you that your manuscript has been deemed suitable for publication in PLOS ONE. Congratulations! Your manuscript is now being handed over to our production team.

Kind regards,

on behalf of

Dr. Muhammad Haroon Stanikzai

Academic Editor

PLOS ONE